# Application of Crustaceans as Ecological Markers for the Assessment of Pollution of Brackish Lakes of Bulgaria Based on Their Ability to Accumulate the Heavy Metals Cd, Zn and Ni

**Elica Valkova** [1,*], **Vasil Atanasov** [1], **Margarita H. Marinova** [1], **Antoaneta Yordanova** [2], **Kristian Yakimov** [1] and **Yordan Kutsarov** [1]

1   Department of Biological Sciences, Agriculture Faculty, Trakia University, 6000 Stara Zagora, Bulgaria; vka@mail.bg (V.A.); mmarinova_@abv.bg (M.H.M.); krisss68@mail.bg (K.Y.); kalimok@gmail.com (Y.K.)
2   Department of "Social Medicine, Health Management and Disaster Medicine", Faculty of Medicine, Trakia University, 6000 Stara Zagora, Bulgaria; antoaneta.yordanova@trakiauni.bg
*   Correspondence: elica_valkova@abv.bg or elitsa.valkova@trakia-uni.bg; Tel.: +359-42-699-314 or +359-889141628

**Abstract:** The present study aimed to assess the pollution of Bulgarian brackish lakes based on their ability to accumulate the heavy metals Cd, Zn and Ni. Physicochemical parameters, including pH, electrical conductivity and salinity of the waters, were determined by potentiometric methods. The heavy metal content of the water and crustacean samples was determined by atomic absorption spectrophotometry. The highest pH in the investigated lakes (Atanasovsko Lake, Poda and Pomorie Lake) in the period May–September 2021 was found in the month of September, in the waters of Atanasovsko Lake (8.84). The concentrations of Cd measured in Atanasovsko Lake in the fall were in the order of 0.0125 µg/L—the highest value recorded for all the studied water bodies. The concentrations of zinc and nickel in the waters did not exceed the norms in Bulgarian legislation. The dynamics of biogenic elements (Zn and Ni) in crustaceans were inversely proportional to those found in the waters. The levels of the toxicant cadmium as well as the metals zinc and nickel in the species *Gammarus* spp. and *Atremia* spp., inhabiting all analyzed water bodies, were significantly lower than those specified in the Bulgarian and European legislation. The pH and electrical conductivity parameters of the tested waters, as well as the concentrations of the heavy metals, cadmium, zinc and nickel, were within the recommended values. High positive correlations were determined between the content of nickel, on the one hand, and cadmium and zinc, on the other, in the organism of the investigated crustaceans. A longer period of research is needed to accurately determine the degree of contamination of these waters.

**Keywords:** pollution; lead; zinc; nickel; accumulation; crustaceans; brackish lakes

## 1. Introduction

Very often, environmental changes are caused by heavy metal pollution. These are elements with a high atomic mass and density above $5 \text{ g/cm}^3$, including lead (Pb), zinc (Zn), mercury (Hg), cadmium (Cd), arsenic (As), nickel (Ni), iron (Fe), copper (Cu), manganese (Mn), etc. [1]. The microelements (Fe, Cu, Zn, Mn, Se, Mo, Cr, etc.) present in natural waters are in the form of organic and inorganic salts. Heavy metals in the water environment can be in three states: dissolved, colloidal dispersed and suspended. In small quantities, some of these metals are necessary for the vital activity and reproduction of all living organisms, performing the role of essential elements. However, above certain values, they have a detrimental effect on the ecological balance and the diversity of hydrobionts [2–5]. Due to their toxic effect, heavy metals disrupt the normal course of biochemical processes in the aquatic inhabitants of natural and cultural ecosystems [6]. These elements, affecting the

hydro-ecosystems, have an impact on the terrestrial ones, as they are spread from them to all trophic levels.

Zinc (Zn) is an important biogenic element that is present in the composition of every cell. Entering the body orally, it is absorbed in the intestines, passes through the liver and is distributed throughout the body. About 90% of the total zinc in the body is concentrated in muscle and bone [7]. The concentration of Zn in natural waters usually does not exceed 0.6 mg/L. Although it is a biogenic element, in high doses, zinc exhibits a strong toxic effect on living organisms, including hydrobionts [8–12]. This element in excessive doses can manifest an estrogenic effect, causing sexual dysfunction, growth and development disorders, and destruction of enzyme systems [13]. In this aspect, zinc, like most heavy metals, is characterized by its ability to accumulate in organs such as the liver, spleen, kidneys and gonads [14–16]. Physiological responses in crustaceans include changes in the activities of major ion pumps and membrane permeability [11,12]. In these hydrobionts, serious lesions have been observed at high levels of zinc ions.

Among heavy metals with high toxicity, the element cadmium (Cd) stands out. This metal is characterized by a normal presence in the environment as an element of soils, air, sediments and even unpolluted sea and fresh waters. Due to the strong toxic effect of Cd, there is substantial information in the literature regarding its impact on living organisms (including hydrobionts). The main accumulation sites of this element in hydrobionts are organs such as the liver and kidneys, which are characterized by the greatest physiological and biochemical activity [17–20]. Studies worldwide have established that Cd induces oxidative stress. This process leads to an increase in lipid peroxidation and the production of reactive oxygen species (ROS) [21–23].

Nickel (Ni) belongs to the heavy elements that have a powerful effect on animal species even at low concentrations [24,25] Exceeding the nickel content is a real danger to hydro-ecosystems due to its resistance and bioaccumulation [26]. Its presence in surplus has a negative impact on the growth, reproduction and behavior of water organisms [27]. The exposition of Ni in the aquatic environment is mainly made by absorption and contact with the skin and gills [28]. Water invertebrates (including crustaceans) are a major nutritional resource for fish and, accordingly, an important connection in the Ni transport chain to the fish [29]. The toxic effects of this metal in fish are well evaluated [30–35], while information on invertebrate organisms is quite limited [36–38]. Numerous studies show that $NI^{2+}$ induces toxicity by inhibition of $Na^+$-$Ca^{2+}$ exchange [39–42]. Under the influence of high levels of nickel, oxidative damage to DNA and proteins in the cell occurs [43].

Water organism's embryos are more sensitive to $Ni^{2+}$ than adult individuals [44,45]. Aquatic organisms absorb heavy metals during all phases of development, mainly during feeding and water intake. Their accumulation takes place mainly in organs, such as the gills of fish and crustaceans, which carry out filtration of the entire amount of water taken in. In typical cases, although to a lesser extent, accumulation is observed in the gonads and eggs of the hydrobionts. In contrast to the gills, minimal amounts of heavy metals are observed in the musculature of crustaceans in most cases [46]. The final effect of the intake is a complete disruption of the metabolism and the occurrence of pathological changes in the organs and systems of the attacked aquatic species. Therefore, monitoring of aquatic ecosystems must necessarily include permanent control of clinically relevant biological markers in aquatic organisms.

Water bodies such as Atanasovsko Lake, Poda and Pomorie Lake, located in the territory of Bulgaria near the Black Sea, have been poorly studied regarding the content of heavy metals in the water and the organisms of crustaceans (Branchiopoda: Anostraca and Malacostraca: Amphipoda) inhabiting these waters.

Freshwater and marine representatives of crustaceans are extremely sensitive to the concentrations of heavy metals in the water bodies that represent their habitat. It is believed that crustaceans can act as first-line indicators for toxicants with severe consequences such as heavy metals due to their ubiquity, large populations and ability to accumulate these elements in their tissues [47,48]. Crustaceans of the classes Branchiopoda and Malacostraca

are common in freshwater and hypersaline lakes throughout the world, including in desert reservoirs and in ice-covered mountain lakes in Antarctica. They swim "upside down" and feed by filtering organic particles from the water or by scraping algae off surfaces [49]. They are a source of food for many birds and fish. They have the ability to accumulate heavy metals, providing a fast and sensitive response in the accumulation process [20]. Therefore, they have the potential to be good indicators for studying the environment with toxic substances of a similar nature [50]. In this regard, the examined representatives of crustaceans from the Branchiopoda: Anostraca and Malacostraca: Amphipoda possess characteristics that make them suitable objects for indication of heavy element pollution in lake ecosystems. This determined the purpose of the present study, which is to assess the pollution of Bulgarian brackish lakes by determining the levels of Cd, Zn and Ni in the waters and crustaceans inhabiting these waters based on their ability to accumulate these metals. Physicochemical parameters including pH, electrical conductivity and salinity of the waters were determined.

## 2. Materials and Methods

### 2.1. Sampling Points

2.1.1. Atanasovsko Lake Maintained Reserve

Atanasovsko Lake (Ladzha, Burgas saltpans) (Figure 1) is characterized by coordinates: 420°34′00″ N and 270°28′00″ E, UTM grid: NH 31, NH 30. It is located in close proximity to the Black Sea coast, 4 km north of the city of Burgas, Burgas region, Bulgaria. It has the status of a managed reserve, Ramsar site, Global Site of Ornithological Importance (GIBA), Zone A of the Ministry of Health. The area of the reserve is 1002.3 ha, with a buffer zone of around 900 ha; it is an exclusive state property. The volume of the lake is 3.2 million m³, the average depth is 0.30 m, and the water area occupies 10.9 km² [51].

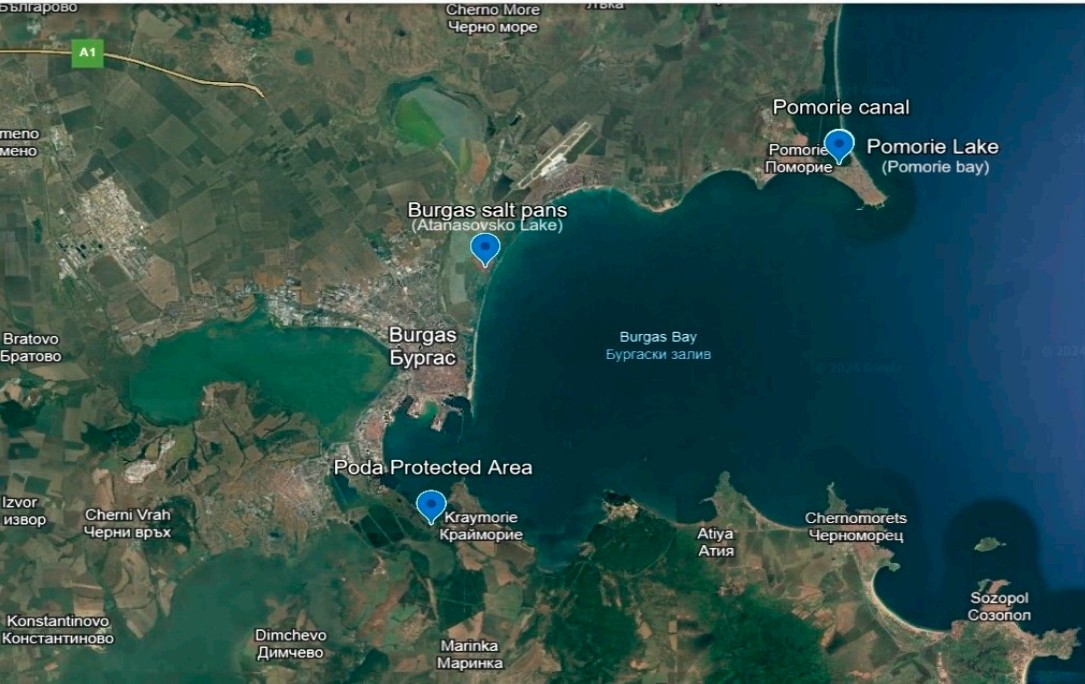

**Figure 1.** Map of the studied area, including the sampling points located near the Bulgarian coast of the Black Sea.

2.1.2. "Poda" Protected Area

The protected area "Poda" (Figure 1) is located on the coast of the Black Sea next to the southern industrial zone of the city of Burgas, Bulgaria. It is part of the complex of the Burgas wetlands, consisting of three large lakes: Atanasovsko, Burgas (or Vaya) and

Mandrensko. It is the easternmost lagoon part of Mandre Lake. To the east it borders the Black Sea (the seashore and the border of swamp vegetation), to the west—the Burgas—Sozopol highway, to the north—the fence of the Iliya Boyadzhiev Shipbuilding and Ship Repair Plant, and to the south—the channel connecting Mandren Lake with the sea. The geographical coordinates of the locality are 27°27′00″ E; 42°27′30″ N. Protected Area (PA) "Poda" is a marshy wetland with an area of 100.7 ha. It has had the status of a Protected Area since 20 April 1989, an Ornithologically Important Place since 1989, and a KORINE place since 1994. It is owned by the Republic of Bulgaria. Protected area "Poda" is managed by the Bulgarian Society for the Protection of Birds (BSPS) [52].

### 2.1.3. Pomorie Lake Protected Area

The Pomorie Lake protected area covers the entire Pomorie Lake (Figure 1). The protected area includes the entire Pomorie Lake wetland and has an area of 760.83 ha. It was declared on 23 January 2001, with the aim of protecting Lake Pomorie, the salt flats and the adjacent coastal areas as a wetland of international importance and as a habitat for 63 species of birds threatened with extinction. The territory of the protected area overlaps with the Pomorie Lake protected area from Natura 2000. In the water area of the protected area, there are also areas from the Ramsar site, the protected areas "Pomorie Lake" and "Pomorie".

The protected area "Pomorie Lake", with identification code BG0000152, is located in the lands of the town of Pomorie and the town of Aheloy, Pomorie municipality, Burgas region, with a total area of 9,215,280 decares, of which 1233 km$^2$ are marine areas. This protected area covers to a large extent both the Pomorie Lake wetland and the eponymous protected area and protected area under the Habitats Directive [53].

The studied lakes have different salinities due to their different infrastructure and connectivity with the Black Sea. Because Atanasovsko Lake is located at a higher altitude than the Black Sea, it is not filled by the sea, but on the contrary, its waters flow to the sea. While the Poda Lake has several channel connections with the Black Sea and a continuous exchange is observed between the two waterbodies.

The studied lakes are located near the Black Sea, but are separated from it by sandbars.

### 2.2. Sampling, Archiving and Storage of Samples

Sampling, archiving and storage of the water samples and samples of crustaceans (Branchiopoda: Anostraca and Malacostraca: Amphipoda) taken from the water bodies in the Atanasovsko Lake Maintained Reserve, the Poda Protected Area and the Pomorie Lake Protected Area were carried out in the period: May 2021–September 2021.

Every month, 3 water samples were taken from each water body and stored for the purpose of analysis according to the requirements of BSS EN ISO 5667-1/2007 [54]; BSS ISO 5667-10:2020 [55]. Sampling points were selected based on a sufficient depth of up to 0.5 m suitable for sampling water and crustaceans. Approximately 4 L of water samples was collected manually in glass containers using personal protective equipment for physicochemical analysis.

Six samples were collected every month from the respective genera *Gammarus* spp. and *Artemia* spp. from studied water bodies. Artemias were only available in Atanasovsko Lake. The identification of the genera was done according to the determiners of Grintsov and Sezgin, 2011 [56] and Timms, 2012 [57]. Crustacean samples were placed in plastic bags, which were transported in ice to the laboratories for study.

### 2.3. Sample Preservation and Analysis

The preservation of the water samples was carried out according to the requirements of BDS EN ISO 5667-3/2012 [58]. The preparation of crab samples was carried out by wet mineralization. Each sample was weighed to a weight of about 1 g and subjected to mineralization in a microwave oven in a mixture of nitric and hydrochloric acids.

The physicochemical parameters pH, electrical conductivity and salinity of the waters were determined by potentiometric methods using electrodes of a combined apparatus.

The analysis of the content of some heavy metals (Zn, Ni and Cd) in the waters of the Atanasovsko Lake Maintained Reserve, Poda Protected Area and Pomorie Lake Protected Area and the organisms of crustacean representatives (Branchiopoda: Anostraca and Malacostraca: Amphipoda), inhabiting these waters was carried out by means of a modern method for the determination of macro and trace elements with an atomic absorption spectrophotometer according to ISO 11047 [59] on a cuvette and flame system using acetylene–oxygen combustion. The heavy metal content of the water and crustacean samples was determined by photometry on a Perkin Elmer AAS 800 atomic absorption spectrophotometer.

### 2.4. Statistical Analyses

Statistical analysis was performed using IBM SPSS 26 for Windows. Statistical differences between indicator values for different lakes were assessed using the Independent-Samples Kruskal–Wallis Test and the Independent-Samples Median Test at Alpha = 0.05. Significance values were adjusted using the Bonferroni correction for multiple testing. Correlations were examined using the non-parametric Spearman rho coefficient, due to the non-normal distribution of some of the variables.

## 3. Results

Determining the physicochemical parameters of water is an important part of monitoring aquatic ecosystems. In the present study, the values of pH, electrical conductivity and salinity of brackish waters characteristic of the studied lakes were measured.

### 3.1. Levels of Physicochemical Parameters pH, Electrical Conductivity (mS/cm$^{-1}$) and Salinity (‰) of the Waters of the Atanasovsko Lake Maintained Reserve, Poda Protected Area and Pomorie Lake Protected Area during the Period May 2021–September 2021

The concentration of H$^+$ has a direct effect on the hydrobionts, such as the studied crustacean species. Due to this fact, it is necessary to monitor the acidity values in the waters of the aforementioned reservoirs.

According to the pH indicator, all analyzed samples (Figure 2) do not exceed the requirements of Regulation H-4 [60] for lakes with mesophilic conditions, which, however, belong to the group with lower salinity below 5‰. However, the samples studied by us are from Black Sea coastal lakes, which are characterized by salinity up to 30‰ (medium saline) and above 40‰ (hypersaline), respectively. In the statistical analysis of the pH data with the Median Test, a statistically significant difference of $p \leq 0.05$ ($p = 0.040$) was found when comparing the values of this indicator from the different sampling points, which suggests the reliability of the obtained results. The average values of pH in Atanasovsko Lake that we found were in May—8.36, in July—8.1 and in September—8.84.

The values recorded by us in the samples from the Poda Protected Area for the months of May, July and September ranged from 7.1 to 7.78.

The acidity of the water samples from Pomorie Bay, measured in May 2021, was 6.81, while the values established in July of the same year were of the order of 8.44. The highest pH was recorded in the month of September—8.63.

The concentration of salts in the studied saline and hypersaline lakes affects the life cycle of the crustaceans and reflects the state of these ecosystems in general. The indicators of electrical conductivity and salinity of the waters reflect the content of the mentioned compounds. When comparing the data from the recorded values of electrical conductivity and salinity of the studied waters, a significance level of $p \leq 0.05$ ($p = 0.044$) was established by means of the Kruskal–Wallis Test, which indicates the good reliability of the results.

The electrical conductivity of the investigated medium-saline and hyper-saline lakes in the period of 2021 (Figure 3) was significantly higher than the regulated limit values for lakes with mesophilic conditions due to the presence of a high concentration of salts.

In Regulation № H-4 [60], no restrictions on this indicator are indicated precisely for this reason—the high level of salinity at these points. It was found that the highest electrical conductivity and salinity is Atanasovsko Lake, whose electrical conductivity varies between 64.7 and 82.95 mS/cm$^{-1}$ and salinity 42.49‰ and 54.5‰, respectively. This assigns Atanasovsko Lake to type L10.

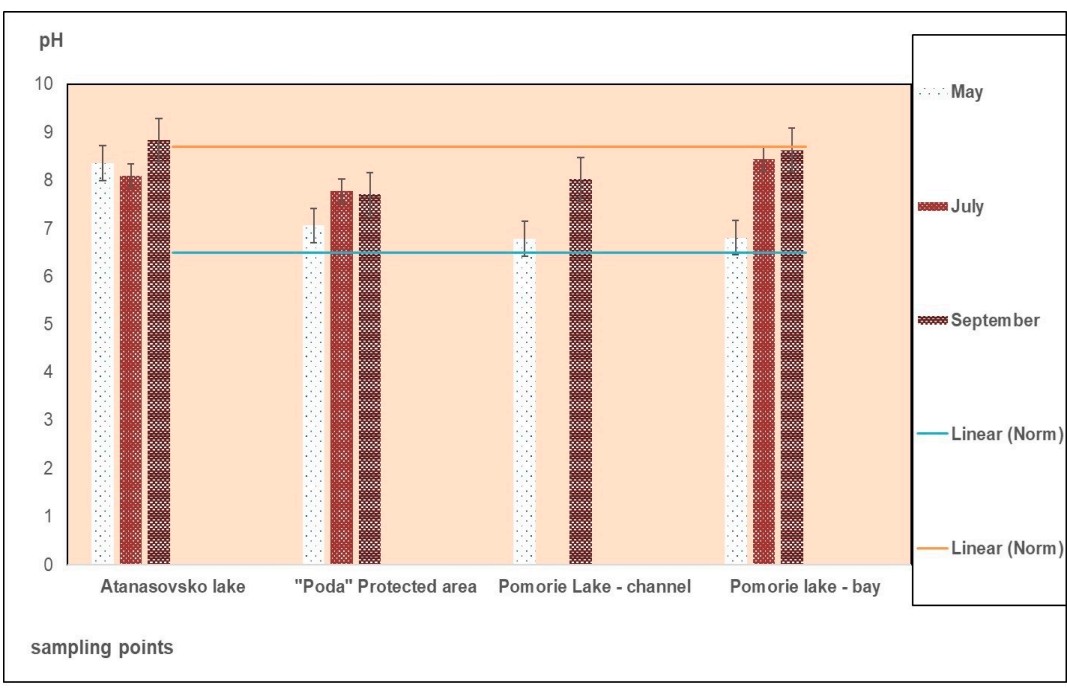

**Figure 2.** pH from the studied sampling points, which belong to type L9 (medium saline) and L10 (extremely saline) lakes. Norm—Regulation № H-4 of 14 September 2012 on the characterization of surface waters from the Bulgarian legislation.

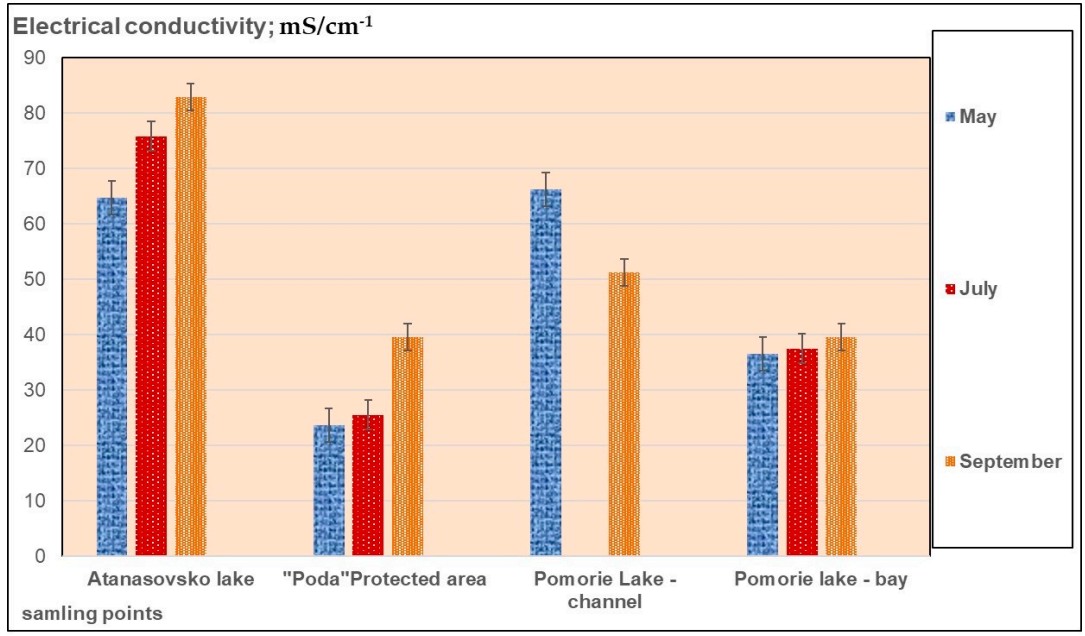

**Figure 3.** Electrical conductivity of the waters from the studied sampling points, which belong to lakes of type L9 (medium saline) and L10 (highly saline) lakes.

From the presented data, it can be seen that there is a tendency (most pronounced at Atanasovsko Lake) to increase water salinity in all water bodies (Figure 4) with the exception of the Pomorie Canal. This is likely due to reduced freshwater inflows during the fall when drought occurs in this geographic area.

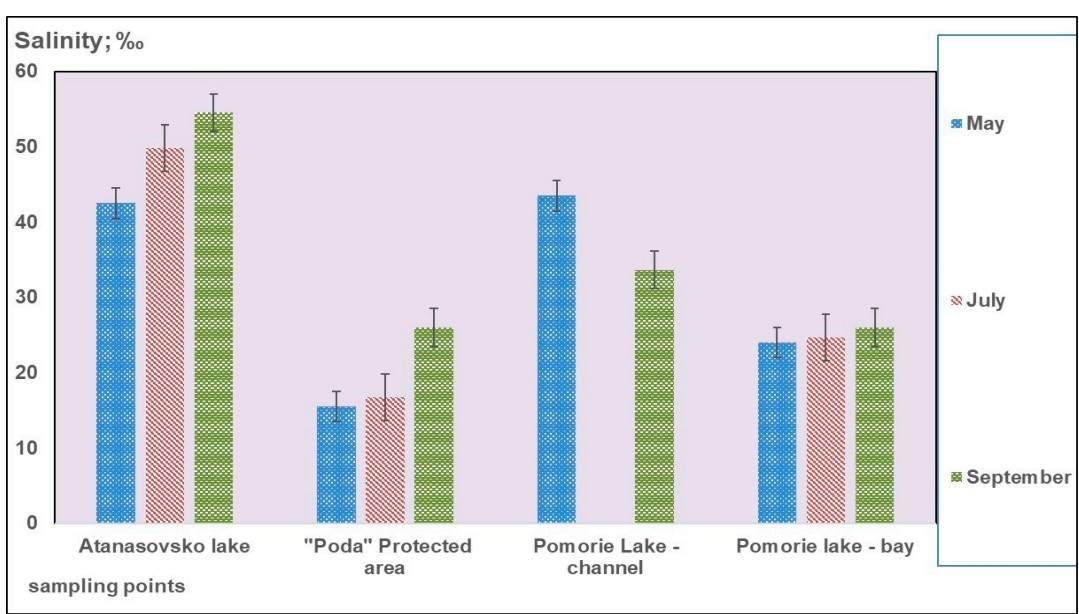

**Figure 4.** Salinity of waters from the investigated sampling points, which fall into type L9 (medium saline) and L10 (extremely saline) lakes.

*3.2. Content of the Heavy Metals Cd, Zn and Ni in the Waters of the Studied Waterbodies of the Atanasovsko Lake Maintained Reserve, Poda Protected Area and Pomorie Lake Protected Area*

The levels of cadmium measured at all the points analyzed by us (Figure 5) were significantly lower than that specified in the Regulation for priority substances and some other pollutants [61]. In this Regulation, the average annual value (AAV–EQS) of cadmium of 0.2 μg/L and the maximum allowable concentration (MAC) of 0.6 μg/L are specified for this type of water. The data analysis of the values of the heavy metal cadmium in the water from the different sampling points using the Kruskal–Wallis test revealed a statistically significant difference of $p \leq 0.05$ ($p = 0.046$), which indicates the good reliability of the results.

The concentrations of Cd measured in Atanasovsko Lake in the autumn were in the order of 0.0125 μg/L, which is actually the highest value recorded for all the studied water bodies. The lowest concentration of 0.0033 μg/L was found in the waters of the Poda Protected Area in May 2021, which is far below the requirements of the Ordinance on priority substances and some other pollutants (0.6 μg/L) [59].

The concentration of cadmium measured in the waters of Pomorie Bay in the period May–September 2021 varied in the range of 0.0052 μg/L–0.0072 μg/L with a trend toward increasing values. Low levels were also recorded in the Pomorie Lake channel in May 2021 (0.0091 μg/L).

All established values of zinc in the water samples from the investigated water bodies (Figure 6) were significantly lower than the AAV–EQS (40 μg/L), as defined in Regulation H-4 [60] for surface waters.

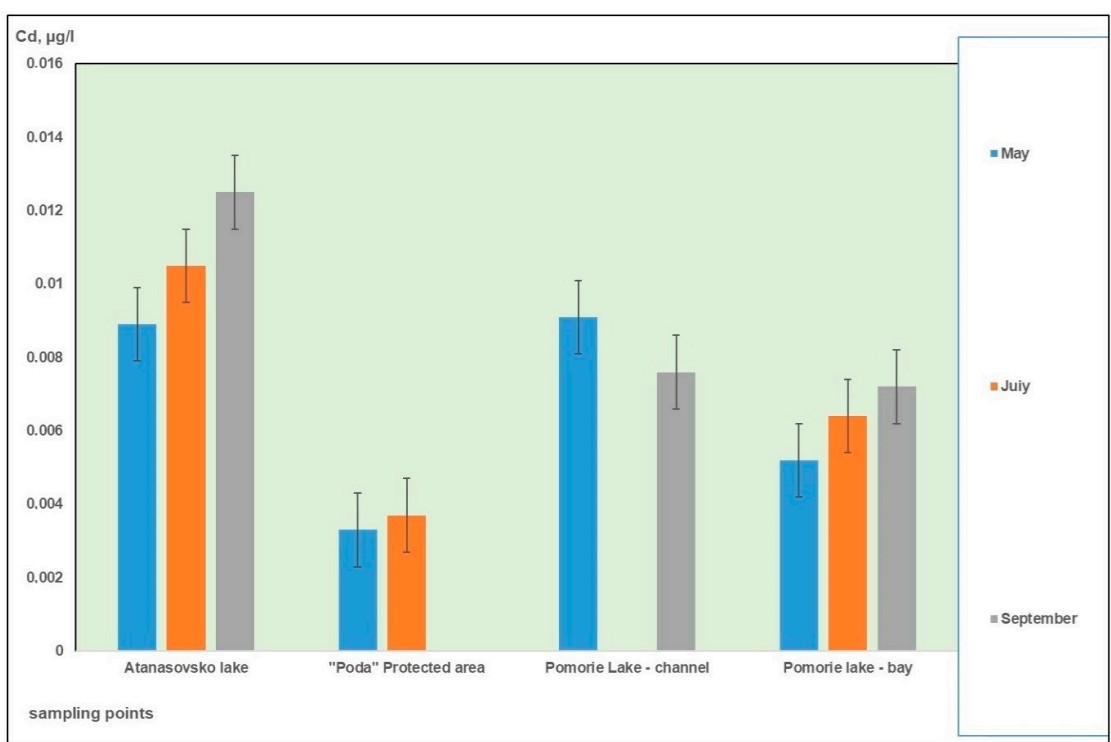

**Figure 5.** Content of Cd in the waters of the studied sampling points, which fall into type L9 (medium saline) and L10 (highly saline) lakes. Norm—Regulation № H-4 of 14 September 2012 on the characterization of surface waters from the Bulgarian legislation.

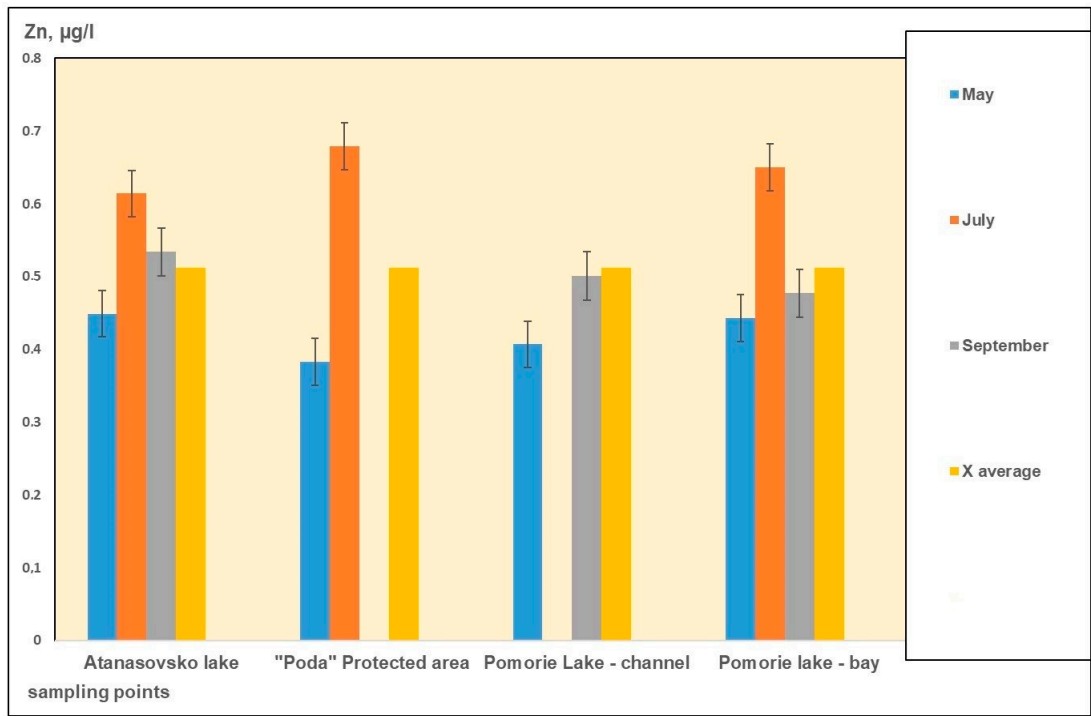

**Figure 6.** Content of Zn in the waters of the investigated sampling points, which belong to type L9 (medium saline) and L10 (extremely saline) lakes.

The levels of Zn registered in the spring season (May 2021—0.449 µg/L) in the waters of Atanasovsko Lake were relatively low compared to the values determined in the month of July (July 2021—0.614 µg/L), when the ambient temperatures were higher, and low

compared to the values established in autumn (0.534 µg/L. The quantities measured in July were higher than X on average for all investigated points (0.512 µg/L).

When determining the quantities of this heavy metal in the waters of the Poda Protected Area, the same phenomenon was observed—the concentrations measured in May (0.383 µg/L) were much lower than those recorded in July (0.679 µg/L. The concentration determined in July was again higher than X on average, which determines an increase in July, but the value was much below the AAV–EQS present in Regulation H-4 [60].

The same tendency of increasing values was also found in the water samples from the Pomorie Lake channel—lower values of Zn in the waters in the spring season (May— 0.407 µg/L and higher in autumn (September—0.501 µg/L).

The concentrations of this metal measured in the waters of Pomorie Bay in May 2021 (0.443 µg/L were lower than those from July of the same year (0.650 µg/L), and this phenomenon was also found at the other points. The levels determined in autumn were lower than in summer (September—0.477 µg/L. Only the concentration of Zn found in summer exceeds the X average for the entire period (0.511 µg/L).

Unfortunately, the described trend regarding the zinc values in the different seasons in the waters from the indicated points was not proven statistically ($p = 0.09$).

The levels of Ni (Figure 7) recorded in all water samples were much lower compared to MAC-AQS, specified in the Regulation on environmental quality standards for priority substances and some other pollutants [61] (34 µg/L). However, a large part of these concentrations exceeds the X average for this metal, determined based on all measured values during the studied period (0.171 µg/L).

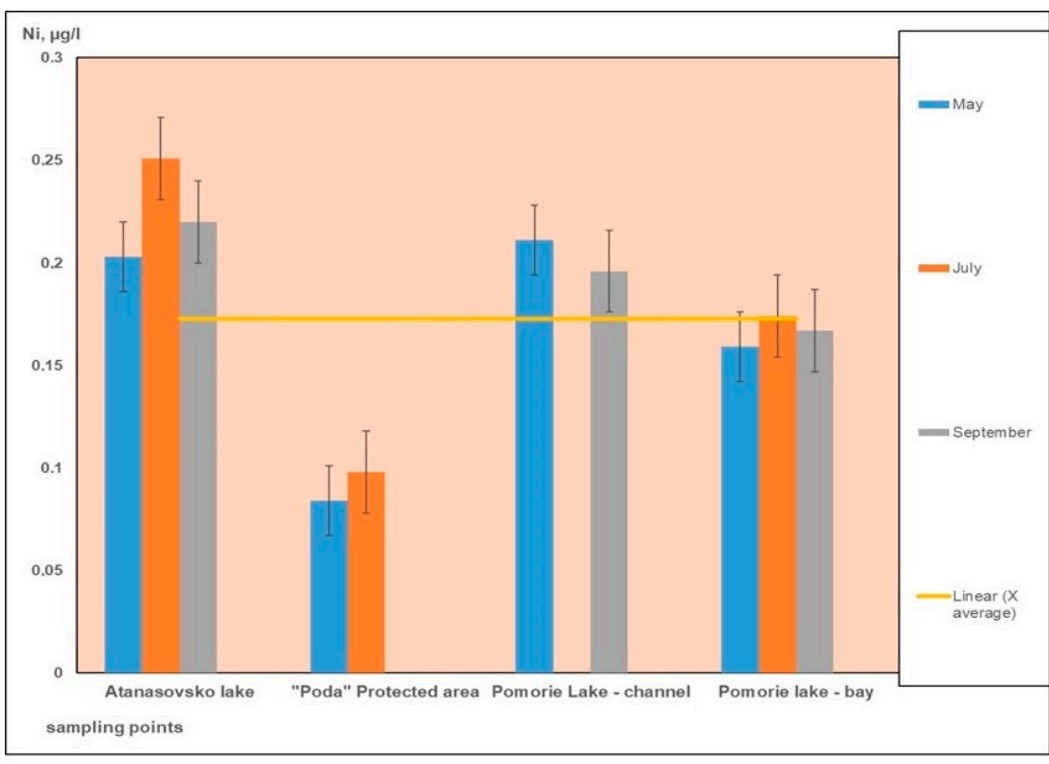

**Figure 7.** Content of Ni in the waters of the investigated sampling points, which fall into type L9 (medium saline) and L10 (highly saline) lakes.

The nickel determined in the water samples from Atanasovsko Lake had lower values in May 2021 (0.203 µg/L) compared to measurements in July (0.251 µg/L) and September (0.220 µg/L) of the same year.

Samples from the Poda Protected Area were found to contain very low concentrations of Ni, with levels in spring (0.084 µg/L) slightly lower than those in summer (0.098 µg/L). Both measurements are below the X average for the period.

The waters in the channel of Pomorie Lake are characterized by lower nickel values in autumn (0.196 μg/L) compared to spring (0.211 μg/L). Both concentrations are higher than the X average, but far from the AAV–AQS specified in the legislation.

The already established tendency for nickel levels is also observed in the concentrations measured in Pomorie Bay (May—0.159 μg/L, July—0.174 μg/Land September—0.167 μg/L), namely, lower values in spring, higher in summer and slightly lowered in autumn.

No statistically significant difference was observed regarding nickel values in the different seasons in the waters from the indicated points ($p$ = 0.12).

*3.3. Content of the Heavy Metals Cd, Zn and Ni in the Organism of Representatives of Crustaceans (Branchiopoda: Anostraca and Malacostraca: Amphipoda) Taken from the Atanasovsko Lake Maintained Reserve, Poda Protected Area and Pomorie Lake Protected Area*

Figure 8 presents the results regarding the content of Cd in the organism of the representatives of crustaceans (Branchiopoda: Anostraca and Malacostraca: Amphipoda) taken from the Atanasovsko Lake Maintained Reserve, Poda Protected Area and Pomoriysko Lake Protected Area. The levels of cadmium found in the body of the investigated crustaceans inhabiting all the analyzed water bodies were significantly lower than that specified in Regulation № 5 of the Bulgarian legislation [62] and Regulation 1881 of the European Union [63] MAC for cadmium content in food (0.5 mg/kg).

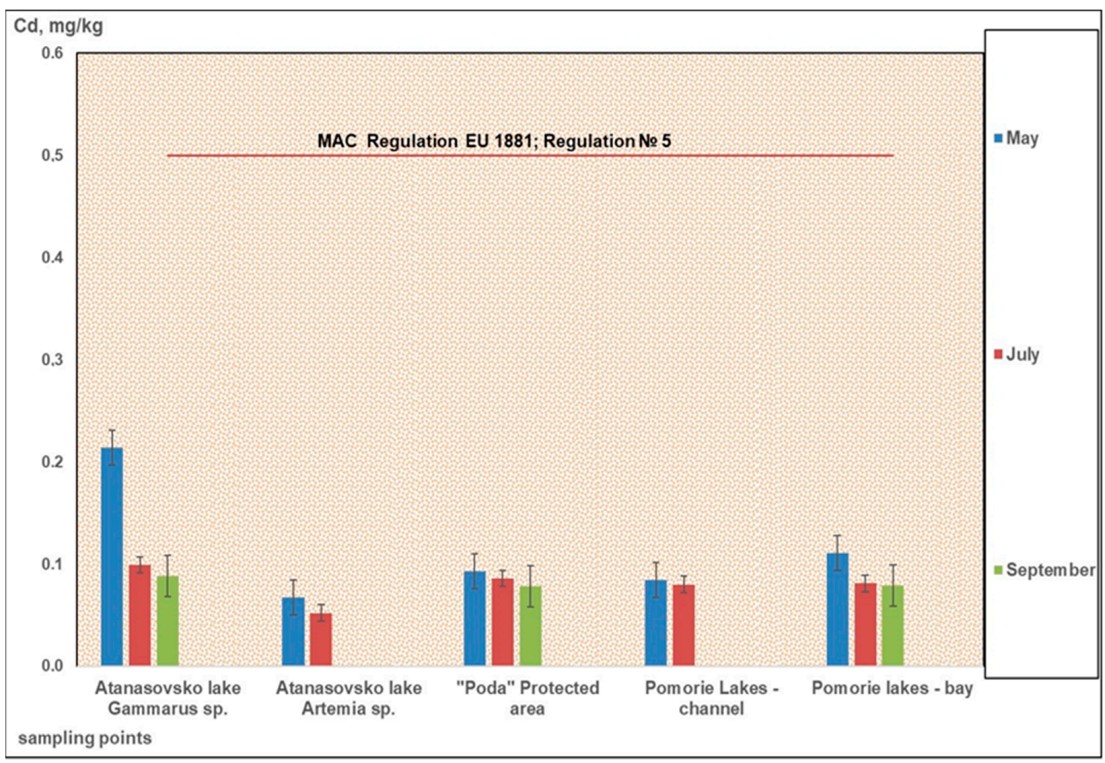

**Figure 8.** Content of Cd in the organism of representatives of crustaceans (Branchiopoda: Anostraca and Malacostraca: Amphipoda) taken from the Atanasovsko Lake Maintained Reserve, Poda Protected Area and Pomorie Lake Protected Area.

The cadmium measured in the samples of *Gammarus* spp., taken from Atanasovsko Lake in the spring season, is characterized by the highest values, which are in the order of 0.214 mg/kg. Significantly lower were the concentrations of this metal reported in samples of *Artemia* spp. taken at the same time and from the same site (0.067 mg/kg), and this was also the lowest value recorded in samples from all water bodies in the spring. When comparing the values of cadmium between Gammarus and Artemia samples, a statistically

significant difference of $p \leq 0.05$ ($p = 0.048$) was found, which confirms the credibility of this statement.

During the summer season, a decrease in Cd levels was observed in all measured samples compared to those recorded in the spring, with the highest values recorded in the organism of the Gammarus inhabiting Atanasovsko Lake. The quantities determined in the Artemia of the same period at this point are significantly lower (0.052 mg/kg) than those measured in crustaceans from all other water bodies.

The same trend was observed in the Cd values recorded in the samples of Gammarus taken from the Poda Protected Area, namely, a decrease in the levels from the spring to the autumn season (0.093 mg/kg–0.078 mg/kg).

In the analysis of Gammarus samples from the channel of Pomorie Lake, a slight decrease in the values from the spring season to the summer (0.084 mg/kg–0.080 mg/kg) was again established. The same statement is valid for the levels of this heavy metal determined in the Gammarus samples from Pomorie Bay of the Black Sea.

The comparison of nickel amounts (Figure 9) will be made based on this concentration due to the lack of specified limits in the food regulations in Bulgaria and the European Union. The average calculated Ni value in the crustacean samples for all measurements made in May, July and September is 0.482 mg/kg.

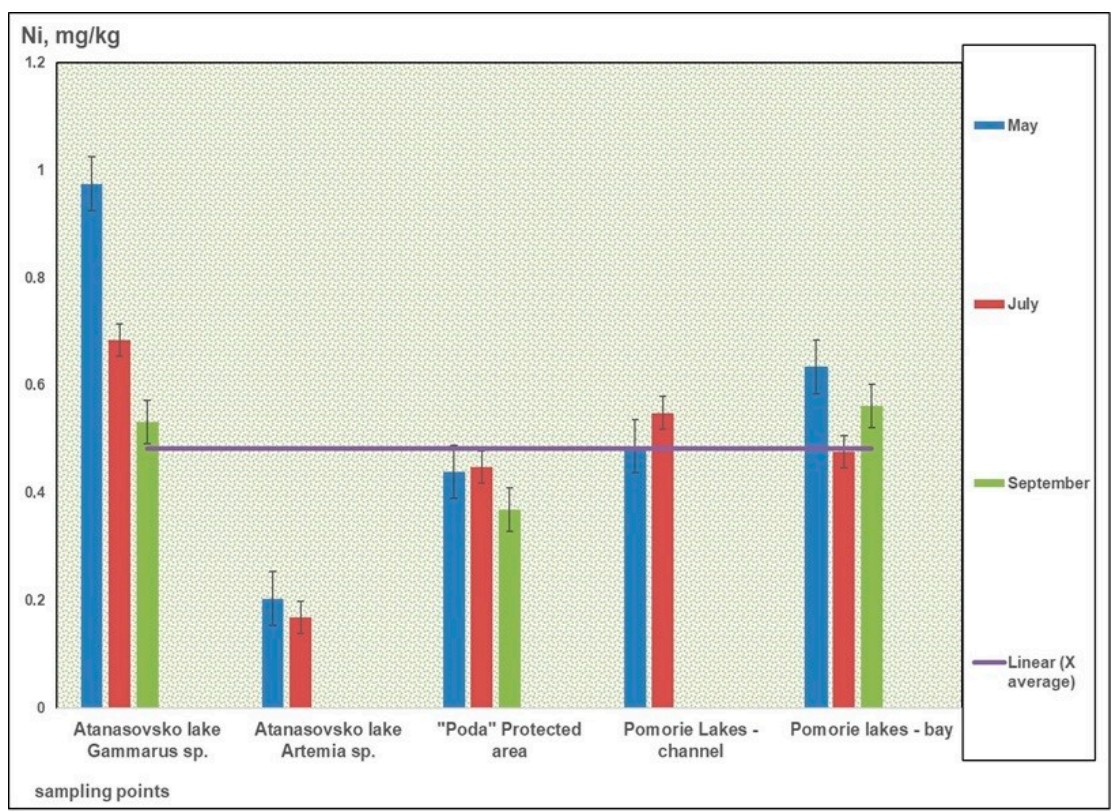

**Figure 9.** Content of Ni in the organism of representatives of crustaceans (Branchiopoda: Anostraca and Malacostraca: Amphipoda) sampled from the Atanasovsko Lake Nature Reserve, Poda Protected Area and Pomorie Lake Protected Area.

Nickel detected in Gammarus samples from Atanasovsko Lake had higher values in May 2021 (0.975 mg/kg) compared to measurements in July (0.685 mg/kg) and September (0.532 mg/kg) of the same year. Ni concentrations in Artemia from Atanasovsko Lake were relatively low compared to those recorded in Gammarus from the same reservoir (May—0.203 mg/kg and July—0.168 mg/kg). The obtained results of a comparison of the values of this metal in the samples of Gammarus and Artemia show the presence of a

statistically significant difference $p \leq 0.01$ ($p = 0.008$), which confirms the credibility of this statement.

Samples from the Poda Protected Area wetland contained lower Ni concentrations in spring (0.439 mg/kg) than in summer (0.448 mg/kg). Both measurements were below the X average for all tests performed. Gammarus from the Pomorie Lake channel is characterized by lower values in spring (0.487 mg/kg) compared to summer (0.549 mg/kg). Both concentrations were higher than the X average for all measurements. Nickel levels in the Pomorie Bay Gammarus samples showed a decrease from May to July (0.634 mg/kg–0.476 mg/kg), followed by a slight increase in September (0.562 mg/kg).

Due to the lack of established norms in Bulgarian and European legislation for the content of the biogenic element Zn in food, the results (Figure 10) will be compared with the calculated arithmetic mean of the total value of this element in all tested crustacean samples in all seasons. This value for the 2021 samples is 22.37 mg/kg.

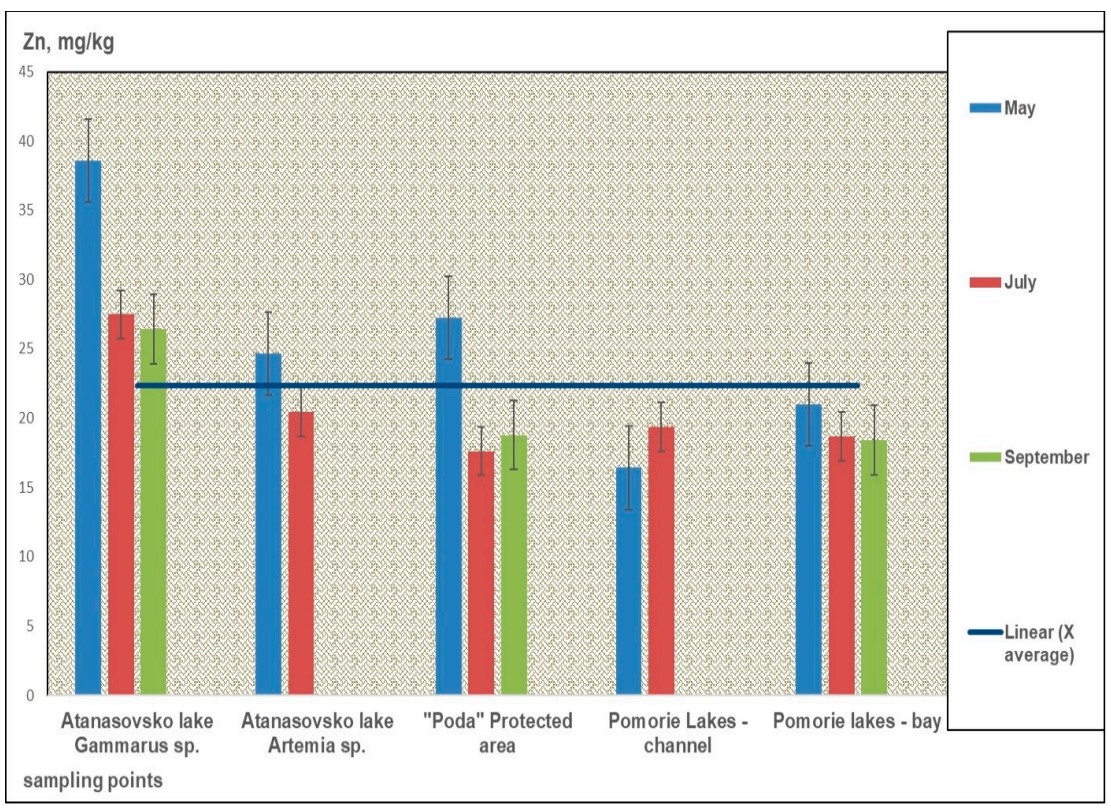

**Figure 10.** Zn content of crustaceans (Branchiopoda: Anostraca and Malacostraca: Amphipoda) sampled from the Atanasovsko Lake Maintained Reserve, Poda Protected Area and Pomorie Lake Protected Area.

The zinc determined in the Gammarus samples from Atanasovsko Lake during the spring season (May 2021—38.6 mg/kg) was much higher than the values measured in July (July 2021—27.48 mg/kg). The concentrations measured in May and July are higher than the X cp for all detected values of this metal.

Analysis of Artemia samples taken in the spring and summer of 2021 showed the same trend, namely, a decrease in values from May to July (from 24.67 mg/kg to 20.45 mg/kg).

When measuring the concentrations of this heavy metal in the Gammarus from the Poda Protected Area, there were fluctuations in the concentrations measured in May (27.22 mg/kg), July (17.62 mg/kg) and September (18.77 mg/kg). Most probably, there is a decrease in zinc consumption by living organisms, which decreases from spring to summer season. The concentration measured in May is higher than that found in July, after which there is a slight increase in autumn due to a decrease in temperatures.

The levels of this metal detected in crustacean samples from Pomorie Bay in May 2021 (20.97 mg/kg) were higher than those in July (18.67 mg/kg) and September (18.39 mg/kg) of the same year.

Unfortunately, when comparing values of this metal by seasons and sampling points, no statistically significant difference was found ($p = 0.1$).

## 4. Discussion

The values of the physicochemical parameters of water are of essential importance for the ecological condition of hydro-ecosystems. The levels of the indicators pH, electrical conductivity and salinity measured in all studied water bodies during the study period did not exceed the legislative norms. The pH values determined by us in Atanasovsko Lake in May (8.36), July (8.1) and September (8.84) were close to the average data published in the MANAGEMENT PLAN OF THE MAINTENANCE RESERVE "ATANASOVSKO LAKE", 2003 [64] (respectively 8.2 and 8.5), and also in the Annual report on the conservation status of habitat 1150 coastal lagoons in BG0000270 Atanasovsko Lake (2021) [65]. According to the research data of Rabadjieva et al. (2018) [66], the acidity of the water in the Poda Protected Area was 8.34, which indicates the presence of a slightly alkaline reaction in this period [67], both according to this study and according to our data. The pH values in the water samples taken for analysis from Pomorie Lake were compared with the data of Hibaum (2010) [68]. The acidity of the samples recorded in the month of May (6.81) was slightly lower than the value found by Hibaum of 8.17 in the same month of study. The data measured in the month of July (8.44) were almost equal to those determined by Hibaum (8.3), while in the month of September, the values found by us (8.63) were higher than those reported by Hibaum (8.15) [68]. There is a tendency to alkalize the waters from spring to autumn, and in September, the waters were alkalized by a further 26%. This alkalization is probably due to a reduced inflow of fresh water as a result of reduced precipitation and an increase in the concentration of alkali and alkaline earth metal salts.

According to the data specified in the project for updating the management plan of the maintained reserve Atanasovsko Lake, in 2014, electrical conductivity varied between 29.6 mS/cm$^{-1}$ and 43.6 mS/cm$^{-1}$, and salinity from 18.1‰ to 27.9‰ [67]. The values obtained during this period refer to Atanasovsko Lake as an L9 type. The data indicated for 2013 in the same document define this basin as an L10 lake type. As previously mentioned, our 2021 data correspond to salinities corresponding to type L10. The established values of electrical conductivity measured in the samples from the Poda Protected area during the period of our study in the months of May, July and September 2021 were 22.5 mS/cm$^{-1}$, 25.5 mS/cm$^{-1}$ and 39.6 mS/cm$^{-1}$, respectively, and salinity—14.77‰, 16.75‰ and 26‰. These results define Poda Lake as an L9-type lake. According to our data, the values of electrical conductivity in the samples from the channel of Pomorie Lake in the months of May and September were respectively 66.26 mS/cm$^{-1}$ and 51.26 mS/cm$^{-1}$, which correspond to salinity 43.5‰ and 33. 6‰. The values we obtained define this point as type L10. At the other point of Pomorie Lake, analyzed by us, the values determined in the months of May, July and September were 36.6 mS/cm$^{-1}$, 37.5 mS/cm$^{-1}$ and 39.6 mS/cm$^{-1}$, which corresponds to salinities 24.04‰, 24.63‰ and 26‰. At these points, the results correspond to L9.

According to the data from a study by Hibaum (2010) [68], the salinity of the Pomorie Lake channel in May was 44.3‰, and this value is close to that obtained in our study. The salinity determined in the late spring was higher than that in the autumn season.

One of the essential hydrochemical indicators is the concentration of heavy metals in the waters. Their levels are a mirror of the ecological–biochemical condition of the studied water bodies [69]. Our results confirm research conducted in the same water bodies from previous years. Both in our study in 2021 in the waters of the Poda Watershed and in the study of Rabadjieva et al. (2020) [70], low levels of cadmium were found in the same water body in June 2015 (0.005 μg/L). The values of this element in the present study represent only 0.55% of the permissible MAC (0.6 μg/L), while the amounts reported in

2015 were 0.83% of the norm. In the study by Rabadjieva et al. (2020) [70], conducted in 2015, similar values of cadmium were found in the waters of Pomorie Lake in the month of June (0.01 µg/L), which are close to the data we found from the end of May 2021 (0.009 µg/L). The levels of this heavy metal determined in the water analysis of May 2021 represent 1.5% of the MAC, and the values reported in the study of Rabadjieva et al. from 2015 were 1.6% of the norm. The Cd values recorded by us at the points of the Pomorie Lake, as well as those cited by Hibaum (2010) [68], are significantly lower than the norms defined in the legislation regarding AAV–EQS (Environmental quality standards) and MAC–EQS for this type of water (0.2 µg/L and 0.6 µg/L). The amounts of Cd determined in our study represent 1.5% of the maximum permissible norms.

Correlation analysis of data from the present study on water cadmium values and electrical conductivity and salinity confirms the high level of positive correlation between these parameters (R = 1).

Zinc, due to its role as a biogenic trace element in living organisms, does not have a toxic effect even at high doses. Due to this fact, in the regulatory documents for this type of water, the set limit regarding the average annual values is high—AAV–EQS (40 µg/L). There are no regulatory restrictions on the maximum allowable concentration of zinc in water. The obtained data regarding the concentrations of this metal in the waters of the investigated reservoirs in this study have a degree of credibility $p \leq 0.05$.

The zinc levels in the summer season recorded in the waters of Atanasovsko Lake were 38% higher than those measured in May 2021. This is completely understandable, since the concentration of salts (salinity) in July was in the order of 49.8‰, and in May—42.49‰.

The phenomenon observed in the waters of the Poda Protected Area, namely, high values of Zn in July compared to May, is due to an increase in the consumption of zinc by living organisms, which intensifies from the spring to summer season, and also to higher temperatures in summer, which leads to a decrease in water levels. Salt levels in July 2021 were higher compared to May of the same year by 5%, which also explains the results obtained. In the study by Rabadjieva et al., 2018 [66], traces of zinc (<0.005 µg/L) were found in the waters of the Poda Protected Area at the end of the spring season. The observed increase in zinc values in the channel of Pomorie Lake from May to September is insignificant, and this is also proven by the salinity values (33.63%). Fluctuations in zinc levels in the waters of Pomorie Bay from the spring to the autumn season are most likely explained by the optimization of temperatures and, accordingly, the metabolism of the hydrobionts consuming this biogenic element. According to Hibaum (2010) [68], the available amounts of zinc in the waters of Pomorie Lake are below the detectable minimum and far below the regulatory requirements. In our study, higher zinc values were found in the waters of Pomorie Bay in May 2021 (0.4 µg/L) compared to those found by Rabadjieva et al., 2020 [70], in June 2015. (0.2 µg/L).

Nickel, which is a trace element with an essential role in the organism of hydrobionts and, in high doses, in water, does not cause intoxication.

The same trend is observed for Ni as for the other studied metals in the waters of Atanasovsko Lake, namely, the values measured in the summer of 2021 were 23% higher than those found in the spring season and 12% higher compared to those measured in autumn of the same year. The quantities of this metal in the month of May 2021 in the samples of the Poda Protected Area were lower by 16% compared to those measured in the month of July, which confirms the established dependence. Apparently, due to the increasing salinity in the summer, there is an increase in the nickel levels in the waters of the investigated reservoirs. The obtained correlation coefficient proves this tendency (R = 0.827). Rabadjieva et al. (2018) [66] also recorded traces of nickel in June 2015 in their study (0.005 µg/L). The results regarding nickel in the waters of Pomorie Bay are no different, where the values registered in July 2021 were 9.43% higher than in May 2021. In autumn, nickel levels were 5% higher than in spring, but also lower than in summer (4%).

The concentration of heavy metals in the body of bioindicators, such as crustaceans of the species *Gammarus* spp. and *Artemia* spp. gives a clearer picture of the actual condition

of the studied water bodies, since these hydrobionts have the ability to accumulate the indicated elements over time, giving information about a past period. The heavy metal cadmium, even in small doses, is a toxicant for all aquabiotics, and this metal has the ability to accumulate in their bodies. Many types of hydrobionts, due to their high sensitivity, represent excellent bioindicators showing the state of the hydro-ecosystem and registering even temporary old peak pollution [7,71,72]. This is why the objects of our study are the crustaceans of (Branchiopoda: Anostraca and Malacostraca: Amphipoda).

The results of our study show a tendency toward a decrease in the cadmium values in the body of the examined Gammarus and Artemia inhabiting all the studied reservoirs from the spring season to autumn. This fact can be explained by the fact that in the spring, the physicochemical parameters of the water environment (including the temperature) are optimal and favor an optimal speed of metabolic processes, while these indicators gradually deteriorate in the summer and autumn [73].

The levels of Cd measured in the Gammarus samples that inhabited Atanasovsko Lake in May 2021 were 78.69% higher than the recorded amounts in Artemia from the same period ($p \leq 0.01$) and 53.74% higher than the cadmium concentrations in Gammarus taken from the same reservoir in July 2021. The amounts of this metal are 12.5% higher in Gammarus from the summer season compared to those in the autumn samples.

The cadmium detected during the spring season in the body of the Gammarus inhabiting the Poda Protected Area was characterized by a value that is 7.53% higher than that found in the summer and 19.35% higher than the measured concentration in the Gammarus samples taken in autumn. There is a tendency to decrease the levels of this toxicant from spring to autumn.

Unfortunately, there are no published data on the content of these heavy metals in the crustacean organisms (Branchiopoda: Anostraca and Malacostraca: Amphipoda) in the Atanasovsko Lake region, Poda Lake and Pomorie Lake, to be compared with our data. In this regard, our studies are pioneering and make a significant contribution to characterizing the state of the studied ecosystems.

The amounts of Ni recorded in all seasons in the *Gammarus* spp. samples from Atanasovsko Lake are higher compared to the determined mean value of this metal for all samples examined of 0.482 mg/kg. The same tendency toward decreasing nickel levels from spring to autumn was observed [74]. Concentrations of this metal recorded in May 2021 are characterized by higher values of 29.74% compared to those of July and by 45.43% compared to those found in September of the same year. This is probably because of a forced spring metabolism of Gammarus, which gradually declines in autumn.

In the Artemia samples from Atanasovsko Lake, Ni abundances in May 2021 were again higher by 17.24% compared to July. The measured concentrations in this crustacean species were lower than those determined in Gammarus from the same water body, where the result was more pronounced in spring and the difference was 79.18%. This result has a high degree of confidence ($p \leq 0.01$). There was an ability of Gammarus to accumulate this metal several times more intensively than Artemia during the study period.

The nickel levels measured in the crustaceans from the Poda Protected Area were almost unchanged when comparing the values in spring and summer.

Nickel detected in samples from Gammarus inhabiting Pomorie Bay was again characterized by a decrease in values from spring to summer by 24.92% and a decrease from spring to autumn by 11.36%. A real decrease was observed in July, followed by a slight increase in September.

The Zn values recorded in the Gammarus samples inhabiting Atanasovsko Lake during the study period of 2021 were higher than the determined arithmetic mean value for this time interval. There is the same trend of decreasing values of this metal in this crustacean from spring to summer season by 28.31%.

Levels determined in Artemia from Atanasovsko Lake are again characterized by a decrease in values from May to July by 17.11%. It is important to note that the zinc levels measured in the Artemia samples in the spring of 2021 were 36.08% lower than

those found in the Gammarus from the same period. This result supports the thesis that Artemia accumulates toxicants such as heavy metals at a significantly lower rate relative to Gammarus. It is most likely not only the species but also the difference in the size of these hydrobionts that matters [72].

The Zn concentrations that were obtained from the analysis of the Gammarus from the Poda Protected Area that inhabited the pond during the spring season were higher by 35.27% compared to the values recorded during the summer season, and by 31.04% compared to the values found in the autumn. In this case, there was a decrease in values from May to July, followed by a slight increase in September of the same year.

There was a gradual decrease in zinc levels in the Gammarus samples taken from the waters of Pomorie Bay from May to September 2021, with concentrations detected in spring being 10.96% higher than summer samples and 12.30% higher than crustacean samples recorded in the autumn season.

## 5. Conclusions

The levels of the parameters pH, electrical conductivity and salinity of the investigated brackish waters are far below the established normative requirements of the Bulgarian legislation.

The measured concentrations of the toxicant cadmium in the waters of the Atanasovsko Lake, Poda, the Pomorie Lake Canal and Pomorie Bay are characterized by the highest values in the autumn season. The amounts of Cd measured in Atanasovsko Lake in the autumn are of the order of 0.0125 µg/L, which is actually the highest value recorded for all the studied water bodies. The lowest concentration of 0.0033 µg/L was found in the waters of the Poda Protected Area in May 2021. The levels of zinc and nickel in the summer season recorded in the waters of Atanasovsko Lake and Pomorie Bay are higher than those measured in May 2021. The concentrations of these biogenic elements do not exceed the established norms.

The registered levels of Cd in crustaceans (*Gammarus* spp. and *Artemia* spp.) inhabiting all analyzed water bodies are significantly lower than the MAC for Cd in food (0.5 mg/kg). The analysis of crustacean samples from the two studied species inhabiting Atanasovsko Lake during the study period revealed a tendency of decreasing Ni values from the spring to the autumn season. A similar trend is observed with the biogenic element zinc.

The levels of the study heavy metals in the samples of Gammarus from Atanasovsko Lake were higher than those reported in the Artemia that inhabited the lake during the same study periods. This result shows the ability of crustaceans of the genus *Gammarus* spp. to accumulate heavy metals times more than representatives of the genus *Artemia* spp. This makes them better biological markers of environmental pollution with heavy metals.

The levels of physicochemical parameters and metals in the waters of the studied lakes are far below the established normative requirements of Bulgarian legislation. The recorded concentrations of these metals in the organisms of crustaceans of the species *Gammarus* spp. and *Artemia* spp. also do not exceed the regulatory requirements. To determine the actual degree of contamination of the study lakes, the study period should be longer.

**Author Contributions:** Conceptualization, E.V. and V.A.; methodology, E.V. and V.A.; software, E.V.; validation, E.V., V.A., M.H.M., A.Y., K.Y. and Y.K.; formal analysis, M.H.M., A.Y., K.Y. and Y.K.; investigation, M.H.M., A.Y., K.Y. and Y.K.; resources, E.V. and V.A.; data curation, M.H.M., A.Y., K.Y. and Y.K.; writing—original draft preparation, E.V. and V.A.; writing—review and editing, E.V. and V.A.; visualization, E.V.; supervision, E.V. and V.A. project administration, E.V.; funding acquisition, E.V. and V.A. All authors have read and agreed to the published version of the manuscript.

**Funding:** This research was funded by Projects 1. AF/21 "Determination of the ecological and biochemical condition of the Atanasovsko Lake Maintained Reserve, the Poda Nature Conservation Center and the Pomorie Lake Protected Area based on the levels of some hydrophysical and hydrochemical indicators and heavy metals (Cd, Pb, Zn, Cu and Ni) in waters and the body of representatives of crustaceans (Branchiopoda: Anostraca and Malacostraca: Amphipoda)"; 2. Project under the "Scientific Research" Fund, Grant KP-06-N52/9.

**Data Availability Statement:** The data sets presented in the paper are not readily available because the data are part of an ongoing study and the development of a dissertation.

**Conflicts of Interest:** The authors declare no conflicts of interest.

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
