# Peer review of "Application of Crustaceans as Ecological Markers for the Assessment of Pollution of Brackish Lakes of Bulgaria Based on Their Ability to Accumulate the Heavy Metals Cd, Zn and Ni"

_2300-7575, doi:10.3390/limnolrev24030017_

Round 1

Reviewer 1 Report

Comments and Suggestions for Authors

Review for the paper “Physicochemical parameters and content of Cd, Zn, and Ni in water and some crustaceans as ecological markers for pollution assessment of the brackish lakes in Bulgaria” by Elica Valkova, Vasil Kostadinov Atanasov, Margarita Hristova Marinova, Kristian Georgiev Yakimov, Yordan Stoyanov Kutsarov submitted to “Limnological Review”.

The authors conducted a field study to determine the concentrations of heavy metals in three Bulgarian brackish lakes. They found varying levels of pH, conductivity, and salinity, as well as Cd, Zn, and Ni in the water and in aquatic crustaceans. Unfortunately, the paper is descriptive in nature and lacks statistical analysis to support the authors' conclusions. For this reason, substantial revisions are needed to reconsider the paper.

Recommendations.

The abstract requires substantial revision because it contains excessive information, repetition, and questionable claims. For example, L 19-20 and L 26 contain the same data.

L 24-26. "The pH and electrical conductivity parameters of the tested waters ... are also far below the established norms". This statement is hard to understand. A better wording would be "are within the recommended values".

The introduction is too long. The authors should delete excessive information on toxic effects of heavy metals.

Materials and methods. The authors should include a map of the study area in the paper.

More details on sampling should be included. The authors should provide information on the sampling sites within each lake. Preferably, they should indicate them on the map. They should explain the choice of these sampling sites. Sampling depths should be mentioned, as well as the sampling method for crustaceans.

L 209-2011. This text is redundant and should be removed.

The authors use vertical bars to show errors in figures, but they did not specify what type of error they used in the study.

Figure 2. The authors should include the reported limits as they did in Figure 1 or include these values in the text.

Section 3.1. The authors should statistically compare the data presented in this section to examine seasonal patterns in pH, EC, and salinity. In addition, it would be useful to compare these data between sites.

Section 3.2. There are no statistical data to confirm the trends mentioned in the text, which makes these results questionable. The authors should use an appropriate statistical method to compare data with respect to both spatial and seasonal variations.

Section 3.3. Again, statistical analyses are needed to confirm the authors' conclusions.

Also, it is not clear to the reader from the text and figures which taxa were considered by the authors. Did they compare the same taxa or different taxa, or did they use pooled samples? In the latter two cases, they should explain the comparability of the contents in different taxa from different locations.

What species of bivalve molluscs were collected from the lakes?

At present, there is no discernible association between the environmental variables and the concentration of heavy metals in crustaceans. This raises the question of the necessity of environmental data. The authors should employ a pertinent statistical analysis to elucidate the relationships between environmental factors (pH, EC, salinity, and heavy metal content in the water) and the concentrations of heavy metals in crustaceans.

Comments on the Quality of English Language

The English should be revised.

Author Response

Comments and Suggestions from Reviewer 1:

Manuscript ID

limnolrev-3041181

“Application of crustaceans as ecological markers for the assessment of the Bulgarian brackish lakes pollution based on their ability to accumulate the heavy metals Cd, Zn and Ni”

Thank you very much for taking the time to review this manuscript. Please find the detailed responses below and the corresponding corrections and highlighted changes in the re-submitted files.

 Comments 1: The abstract requires substantial revision because it contains excessive information, repetition, and questionable claims. For example, L 19-20 and L 26 contain the same data.

Response 1: Line L26 summarizes the results obtained and should not be removed.

Comments 2: L 24-26. "The pH and electrical conductivity parameters of the tested waters ... are also far below the established norms". This statement is hard to understand. A better wording would be "are within the recommended values".

 Response 2: The text has been edited according to the requirements.

Comments 3: The introduction is too long. The authors should delete excessive information on toxic effects of heavy metals.

 Response 3: The text has been edited according to the requirements.

Comments 4: Materials and methods. The authors should include a map of the study area in the paper.

  Response 4: A map of the study area is included in the paper and sampling points are noted.

Comments 5: More details on sampling should be included. The authors should provide information on the sampling sites within each lake. Preferably, they should indicate them on the map. They should explain the choice of these sampling sites. Sampling depths should be mentioned, as well as the sampling method for crustaceans.

Response 5: On the attached map of the research area, the locations of sampling from the relevant water bodies are marked. Sampling points were selected based on a sufficient depth of up to 0.5 m suitable for sampling water and crustaceans. In the text, the documents from the Bulgarian legislation against which the water and crustacean samples were taken are indicated.

Comments 6:L 209-2011. This text is redundant and should be removed.

Response 6: An introduction is necessary before presenting the results in order to maintain the scientific aspect of the article.

Comments 7: The authors use vertical bars to show errors in figures, but they did not specify what type of error they used in the study.

Response 7: The bars on the graphs indicate the standard deviation of the error of measurement of the respective values.

Comments 8: Figure 2. The authors should include the reported limits as they did in Figure 1 or include these values in the text.

Response 8: For the indicators of electrical conductivity and salinity, no permissible limits are specified in Regulation H-4 of the Bulgarian legislation.

Comments 9: Section 3.1. The authors should statistically compare the data presented in this section to examine seasonal patterns in pH, EC, and salinity. In addition, it would be useful to compare these data between sites.

Response 9: In the text, data from the statistical analysis of the values of the studied physicochemical parameters are indicated.

Comments 10: Section 3.2. There are no statistical data to confirm the trends mentioned in the text, which makes these results questionable. The authors should use an appropriate statistical method to compare data with respect to both spatial and seasonal variations.

Response 10: The text contains data from the statistical analysis of the values of the investigated heavy metals in the waters of the specified water bodies.

Comments 10: Section 3.3. Again, statistical analyses are needed to confirm the authors' conclusions.

Response 10: The text contains data from the statistical analysis of the values ​​of the studied heavy metals in the organism of the crustaceans of the genera Gammarus spp. Amd Artemia spp. that inhabited the studied reservoirs in the specified period.

Comments 11: Also, it is not clear to the reader from the text and figures which taxa were considered by the authors. Did they compare the same taxa or different taxa, or did they use pooled samples? In the latter two cases, they should explain the comparability of the contents in different taxa from different locations.

Response 11: Crustaceans of the genera Gammarus spp. and Atemia spp. were used as objects of study. Artemias were found only in Atanasovsko Lake. The values ​​of heavy metals in the gammarus from the different reservoirs and the values ​​of the metals between the gammarus and artemia from Atanasovsko Lake, taken at the same time in the different seasons, were compared.

Comments 12: What species of bivalve molluscs were collected from the lakes?

Response 12: Mussels were not the subject of our study. Errors in the text have been corrected.

Comments 13: At present, there is no discernible association between the environmental variables and the concentration of heavy metals in crustaceans. This raises the question of the necessity of environmental data. The authors should employ a pertinent statistical analysis to elucidate the relationships between environmental factors (pH, EC, salinity, and heavy metal content in the water) and the concentrations of heavy metals in crustaceans.

Response 13: Our studies are pioneering and no detailed comment has been made on the correlation dependences of physicochemical parameters and heavy metals in aquatic and crustacean. Correlation coefficients are indicated where appropriate.

Reviewer 2 Report

Comments and Suggestions for Authors

Some comments on the paper:
Application of crustaceans as ecological markers for the assessment of the Bulgarian brackish lakes pollution based on 3 their ability to accumulate the heavy metals Cd, Zn and Ni
The authors intend to justify the crustacean species as biomarkers to assess the water pollution with heavy metals. In this context, some data reported at national scale are used relative to Cd, Zn and Ni content of three lakes located near Black Sea coast. The concentration values reported for these heavy metal pollutants are below national and European limits stipulated by regulations in force, therefore no hazardous effect was observed.
The originality and innovative character of this study is not justified enough. Moreover, there are many errors that must be corrected and some paragraphs that need to be reformulated.
The paper could be accepted after these corrections are made.
Replace l by L for unit of liter through the text!
Line 34: “have been” instead of “are have”
Line 35: cm3, not cm3
Line 46: I think that “are” could be placed in front of “spread”
Line 57: “zinc”, not “Zinc”
Line 61: “at” is better than “from”
Line 66: “aquatic systems” or “aquatic flows” not “aquatic ones”
Line 71: to reformulate
Line 76: “even”, not “end”
Line 81: “Ni”, not “NI”
Line 110: “classes”, not “class”
Line 111: without “class” in front of “Malacostraca”
Line 112: “also” is useless after “and”
Line 133: I think that we can add “it is an” in front of “exclusive”
Line 189: what precision?
Observation to Figure 1: In order to put in evidence small variations of pH, it should not start from zero on ordinate axis. It is suggesting the pH interval 5 ÷10.
Lines 300 & 325: without “.” at the end
Lines 324 & 327: a space before 6
Lines 214 & 276 & 346: without “.” at the end of titles
Page 13: There are many not corrected typing errors!
Line 489: with “are”
Lines 509-512: The idea is repeated many times. To reformulate!
Lines 544 & 569 & 599: without “paragraph”.
Line 598: is empty

Comments on the Quality of English Language

The English language needs minor corrections.

Author Response

Comments and Suggestions from Reviewer 2:

Manuscript ID

limnolrev-3041181

“Application of crustaceans as ecological markers for the assessment of the Bulgarian brackish lakes pollution based on their ability to accumulate the heavy metals Cd, Zn and Ni”

Thank you very much for taking the time to review this manuscript. Please find the detailed responses below and the corresponding corrections and highlighted changes in the re-submitted files.

Comments 1: Replace l by L for unit of liter through the text!
Response 1: The text has been corrected as required.

Comments 2: Line 34: “have been” instead of “are have”
Response 2: The text has been corrected as required.

Comments 3: Line 35: cm3, not cm3
Response 3: The text has been corrected as required.

Comments 4: Line 46: I think that “are” could be placed in front of “spread”
Response 4: The text has been corrected as required.

Comments 5: Line 57: “zinc”, not “Zinc”
Response 5: The text has been corrected as required.

Comments 6: Line 61: “at” is better than “from”
Response 6: The text has been corrected as required.

Comments 7: Line 66: “aquatic systems” or “aquatic flows” not “aquatic ones”
Response 7: The text has been corrected as required.

Comments 8: Line 71: to reformulate
Response 8: The text has been corrected as required.

Comments 9: Line 76: “even”, not “end”
Response 9: The text has been corrected as required.

Comments 10: Line 81: “Ni”, not “NI”
Response 10: The text has been corrected as required.

Comments 11: Line 110: “classes”, not “class”
Response 11: The text has been corrected as required.

Comments 12: Line 111: without “class” in front of “Malacostraca”
Response 12: The text has been corrected as required.

Comments 13: Line 112: “also” is useless after “and”
Response 13: The text has been corrected as required.

Comments 14: Line 133: I think that we can add “it is an” in front of “exclusive”
Response 14: The text has been corrected as required.

Comments 15: Line 189: what precision?
Response 15: The text has been corrected as required.

Comments 16: Observation to Figure 1: In order to put in evidence small variations of pH, it should not start from zero on ordinate axis. It is suggesting the pH interval 5 ÷10.
Response 16: In the opinion of the authors, the pH interval remains 0 ÷ 10.

Comments 17: Lines 300 & 325: without “.” at the end
Response 17: The text has been corrected as required.

Comments 18: Lines 324 & 327: a space before 6
Response 18: The text has been corrected as required.

Comments 19: Lines 214 & 276 & 346: without “.” at the end of titles
Response 19: The text has been corrected as required.

Comments 20: Page 13: There are many not corrected typing errors!
Response 20: The text has been corrected as required.

Comments 21: Line 489: with “are”
Response 21: The text has been corrected as required.

Comments 22: Lines 509-512: The idea is repeated many times. To reformulate!
Response 22: The text has been corrected as required.

Comments 23: Lines 544 & 569 & 599: without “paragraph”.
Response 23: The text has been corrected as required.

Comments 24: Line 598: is empty.

Response 24: The text has been corrected as required.

Round 2

Reviewer 1 Report

Comments and Suggestions for Authors

The authors have revised the text but some minor revisions are required as follows:

  1. Check the Latin names and italicize them (Artemia and Gammrus) (for example L. 26).
  2. Change 'artemia' and 'gammarus' with 'Artemia' and 'Gammarus' throughout the text
  3. Change 'have been' with 'were' in Lines 294, 315, 342, 373)
  4. Change 'was varied' with 'varied' (L 472)
  5. Change 'are being' with 'were' (L 467. 478)
  6. All statistical methods used by the authors (KWT, correlation analysis…) should be mentioned in the 'Materials and Methods"
Comments on the Quality of English Language

Revisions are required 

Author Response

Comments and Suggestions from Reviewer 1; Round 2:

Manuscript ID

limnolrev-3041181

“Application of crustaceans as ecological markers for the assessment of the Bulgarian brackish lakes pollution based on their ability to accumulate the heavy metals Cd, Zn and Ni”

Thank you very much for taking the time to review this manuscript. Please find the detailed responses below and the corresponding corrections and highlighted changes in the re-submitted files.

Comments 1: Check the Latin names and italicize them (Artemia and Gammrus) (for example L. 26).

Response 1: The text has been edited according to the requirements.

Comments 2:Change 'artemia' and 'gammarus' with 'Artemia' and 'Gammarus' throughout the text.

Response 2: The text has been edited according to the requirements.

Comments 3: Change 'have been' with 'were' in Lines 294, 315, 342, 373).

Response 3: The text has been edited according to the requirements.

Comments 4: Change 'was varied' with 'varied' (L 472).

Response 4: The text has been edited according to the requirements.

Comments 5: Change 'are being' with 'were' (L 467. 478).

Response 5: The text has been edited according to the requirements.

Comments 6: All statistical methods used by the authors (KWT, correlation analysis…) should be mentioned in the 'Materials and Methods"

Response 6: All statistical methods used in the study are mentioned in “Materials and Methods”.
